# Studies on the Reactions of Biapenem with VIM Metallo β-Lactamases and the Serine β-Lactamase KPC-2

**DOI:** 10.3390/antibiotics11030396

**Published:** 2022-03-16

**Authors:** Anka Lucic, Tika R. Malla, Karina Calvopiña, Catherine L. Tooke, Jürgen Brem, Michael A. McDonough, James Spencer, Christopher J. Schofield

**Affiliations:** 1Chemistry Research Laboratory, The Department of Chemistry and the Ineos Oxford Institute for Antimicrobial Research, University of Oxford, Oxford OX1 3TA, UK; anka.lucic@strubi.ox.ac.uk (A.L.); tika.malla@chem.ox.ac.uk (T.R.M.); karina.calvopinatapia@chem.ox.ac.uk (K.C.); jurgen.brem@chem.ox.ac.uk (J.B.); michael.mcdonough@chem.ox.ac.uk (M.A.M.); 2Biomedical Sciences Building, School of Cellular and Molecular Medicine, Faculty of Life Sciences, University of Bristol, University Walk, Bristol BS8 1TD, UK; ct12425@bristol.ac.uk (C.L.T.); jim.spencer@bristol.ac.uk (J.S.)

**Keywords:** carbapenems, antimicrobial resistance, metallo-β-lactamases, serine-β-lactamases, biapenem

## Abstract

Carbapenems are important antibacterials and are both substrates and inhibitors of some β-lactamases. We report studies on the reaction of the unusual carbapenem biapenem, with the subclass B1 metallo-β-lactamases VIM-1 and VIM-2 and the class A serine-β-lactamase KPC-2. X-ray diffraction studies with VIM-2 crystals treated with biapenem reveal the opening of the β-lactam ring to form a mixture of the (2*S*)-imine and enamine complexed at the active site. NMR studies on the reactions of biapenem with VIM-1, VIM-2, and KPC-2 reveal the formation of hydrolysed enamine and (2*R*)- and (2*S*)-imine products. The combined results support the proposal that SBL/MBL-mediated carbapenem hydrolysis results in a mixture of tautomerizing enamine and (2*R*)- and (2*S*)-imine products, with the thermodynamically favoured (2*S*)-imine being the major observed species over a relatively long-time scale. The results suggest that prolonging the lifetimes of β-lactamase carbapenem complexes by optimising tautomerisation of the nascently formed enamine to the (2*R*)-imine and likely more stable (2*S*)-imine tautomer is of interest in developing improved carbapenems.

## 1. Introduction

β-Lactam-containing drugs are the most clinically important antibacterial class [1], though their use is increasingly impaired by resistance due to nucleophilic serine-β-lactamases (SBLs, Ambler classes A, C and D) and zinc-dependent metallo-β-lactamases (MBLs, Ambler class B) (Figure 1) [2,3,4]. Carbapenems were once considered antibiotics of ‘last resort’, in part because they are resistant to hydrolysis by some SBLs [1]. However, the increasingly widespread use of carbapenems is correlated with an increase in extended-spectrum SBLs (ESBLs) in both Gram-positive and Gram-negative bacteria [5]. The role of MBLs, such as the Verona integron MBLs, which often efficiently hydrolyse carbapenems, in antimicrobial resistance is also rapidly growing and is now endemic in some regions [6].

The stability of the complexes formed by β-lactams with SBLs/MBLs is an important factor in determining their susceptibility to β-lactamases and hence efficacy. The ability of carbapenems to form relatively stable complexes both with their transpeptidase targets and certain SBLs has contributed to their greater efficacy compared to other β-lactams against some resistant strains. However, there is incomplete knowledge of the molecular factors involved in determining enzyme-carbapenem complex stabilities, as exemplified by the recent observation that carbapenems react with Class D SBLs to produce β-lactones in addition to the established enamine and imine hydrolysed products (Figure 1) [7,8]. Studies on the products of carbapenem hydrolysis by several MBLs and SBLs have led to the proposal that, at least in some cases, during efficient catalysis the major nascent product is in the enamine tautomeric form, which undergoes isomerisation to give the (2*R*)- and (2*S*)-C2 imines (Figure 1) [8,9,10,11].

The intravenously administered carbapenem biapenem has a broad spectrum of activity and is used to treat serious infections [12,13,14]. Biapenem is reported to have less adverse side effects than other carbapenems and does not need to be administered with a dihydropeptidase 1 inhibitor [13]. Biapenem is an inhibitor of some SBLs but is subject to MBL hydrolysis [12,13,14]. Since biapenem has the same C6 hydroxyethyl side chain as other carbapenems, its distinctive properties must be due to the presence of its unusual positively charged ‘bicyclotriazolium’ sulfur-linked C2 bicyclic ring system (Figure 1) [12,13,14]. The unusual C2 side chain of biapenem compared to the more typical C2 side chains of carbapenems such as meropenem makes it of particular interest with respect to defining its tautomeric enamine and imine product profile.

Here, we report studies on the reactions of biapenem with the clinically important VIM-1, VIM-2, and KPC-2 β-lactamases using NMR spectroscopy and, in the case of VIM-2, X-ray crystallographic analysis. After the pioneering identification of the first Verona integron MBL (VIM) in 1996, multiple VIM variants have been observed in Gram-negative bacteria, including in *Klebsiella* and *Eneterobacteriaceae* [15,16]. The VIM MBLs are broad-spectrum β-lactamases that catalyse the hydrolysis of penicillins, cephalosporins, and carbapenems, but not monobactams [17,18,19,20] (Figure 1C). VIMs have the typical αβ/βα MBL fold and are di-zinc utilising B1 subfamily MBLs [17,19,20]. The *Klebsiella pneumoniae* carbapenemase 2 (KPC-2) is a Class A SBL of increasing clinical importance and has broad-spectrum activity, including with respect to the particularly efficient hydrolysis of carbapenems [21,22].

## 2. Results

We initially investigated how biapenem reacts with VIM-1, VIM-2, and KPC-2 using ^1^H NMR (600 MHz) spectroscopy. In all three cases, enamine and both (2*R*)-imine and (2*S*)-imine products were observed, as evidenced by the analysis of the methyl group resonances (C6 hydroxyethyl methyl and C1 methyl groups) in the high-field region (0.8–1.3 ppm) of the ^1^H NMR spectrum. The major product observed was the (2*S*)-imine, consistent with previous work indicating that this is the thermodynamically favoured species of the three observed major tautomeric products. [7,8,9,20]. Lower levels of the (2*R*)-imine and enamine products were observed, along with low levels of at least one unassigned product with methyl group resonances at ~1.00 and 1.02 ppm (Figure 2 and Appendix A). No evidence for the formation of lactone products was accrued, contrasting with the results of reaction of carbapenems with class D SBLs, but consistent with prior work on the reactions of carbapenems with MBLs (Figure 1), [7,8,9,20]. Overall, our NMR results support the proposal of the (2*S*)-imine as the major product (on the timescale of the NMR assays), though they do not preclude the possibility that the enamine or (2*R*)-imine are nascent products.

Crystal structures have been reported for biapenem-derived complexes with the class B2 MBL CphA and certain transpeptidases [23,24,25]. However, there is no reported structure for biapenem with a representative of the most clinically relevant class B1 MBLs. Thus, we performed studies to investigate how biapenem interacts with the B1 MBL Verona integron metallo β-lactamase (VIM-2), on which studies with inhibitors have previously been carried out [26,27]. Crystals of recombinant VIM-2 were produced as previously described [20,26] and were soaked with a 50 mM solution of biapenem for 1, 5, 10, and 30 min. A diffraction dataset with a crystal soaked in biapenem for 10 min was obtained with the Diamond Light Source at a 1.3 Å resolution (space group *I*222 with one molecule in the asymmetric unit) (Appendix A). Due to low completeness in the high-resolution shell, the resolution for data processing was cut to 1.5 Å.

Analysis of the electron density at the VIM-2 active site led to trial refinements with either the (2*S*)-imine, the (2*R*)-imine or the enamine (independently), the two imines together at 50% occupancy, or the enamine with either the (2*S*)- or (2*R*)-imine at 50% occupancy (Figure 3 and Figure 4). The best fit was obtained with the (2*S*)-imine and the enamine at 50% occupancy. Note that the occupancy level should be regarded as an estimation and we cannot rule out low levels of the presence of the (2*R*)-imine and other ligands; in the NMR studies, there were some unassigned peaks, consistent with minor product formation (Appendix A).

The apparent electron density for the bicyclic side chains of both the (2*S*)-imine and the enamine ligands was weak, likely due to the lack of protein contacts causing conformational mobility (Figure 3). In general, the interactions made by both the enamine and (2*S*)-imine are analogous to those observed in other VIM-1: carbapenem-derived structures [19], i.e., in both cases the newly formed β-lactam derived carboxylates displace the water/hydroxide ion bridging the two zinc ions, and the C3 carboxylate is positioned to interact with the Arg228 sidechain (Appendix A). An increase of 0.8 Å in the distance between the two active site zinc ions (4.3 Å) compared to the unliganded structure was also observed, as previously reported in MBL carbapenem and faropenem derived complexes [19,20].

Notably, the thioether of the C2 biapenem-derived sidechain adopts different conformations for the enamine and the (2*S*)-imine ligands, with the sulfur being positioned close to the primary amide of Asn233 in the case of the enamine, but not the (2*S*)-imine (Figure 4 and Appendix A). The C6 hydroxyethyl group is positioned similarly for both the enamine and (2*S*)-imine ligands, in a manner similar to that observed in previous VIM-2 carbapenem/faropenem-derived complexes (Appendix A) [19,20].

When compared with a VIM-1: meropenem-derived (2*S*)-imine structure (PDB: 5N5I) [19], the binding modes in the VIM-2: biapenem-derived complex structure manifest some differences (Appendix A). The C6 hydroxyethyl group and newly formed carboxylate in the VIM-1: meropenem (2*S*)-imine product complex structure [19] have different orientations compared to VIM-2: biapenem products complex for both the enamine and the (2*S*)-imine. Notably, the binding modes of the VIM-2: biapenem-derived ligand complex and the VIM-2: faropenem Z-alkene imine-derived ligand complex appear more similar, with the analogous positioning of the C2 carboxylate (Appendix A) [20]. There are also differences in the binding modes of biapenem derived ligands when reacted with penicillin binding proteins and the class B2 MBL CphA (Appendix A). It is also notable that the different tautomeric complexes formed by carbapenem-derived ligands with MBLs are mimicked to different extents by various heterocyclic MBL inhibitors binding at the di-Zn(II) containing active site, as observed by crystallography [28,29].

## 3. Discussion

Previous studies have shown that the SBL/MBL catalysis of carbapenem hydrolysis results in the formation of enamine and (2*R*) and or (2*S*)-imines as the major observed hydrolysed products observed in the NMR analyses, with the analogous β-lactones being additionally observed in the case of class D SBLs [7,8,9,11]. Consistent with these findings, in the results reported here with biapenem, the (2*S*)-imine was observed as the major product with the B1 metallo-β-lactamases VIM-1, VIM-2 and the class A serine-β-lactamase KPC-2, at least on the relatively long timescale of the NMR assays. This observation indicates that the unusual biapenem bicyclotriazolium side chain does not substantially perturb the enamine/imine equilibrium position in the product enamine and imine tautomers (Figure 2 and Appendix A).

We appreciate that care should be taken in assigning crystallographically observed complexes, at least in terms of the details of binding, as being relevant to catalytically relevant intermediate structures in solution. However, consistent with the solution studies employing NMR, the crystallographic analysis of VIM-2 crystals soaked with biapenem revealed evidence for the enamine and the (2*S*)-imine as the major products bound, with no evidence for the (2*R*)-imine. We cannot rule out that the (2*R*)-imine, along with other ligands, may be formed at the active site at low levels, either in crystallo or in solution. In solution, at least for some carbapenems, the (2*R*)-imine is the kinetic product of enamine tautomerisation [7,9,10]. Thus, it is possible that we did not observe the (2*R*)-imine in part because of the relatively long timescale of the crystallographic analyses, which involved a 10 min crystal-soaking procedure. However, our combined results are also consistent with the proposal that during efficient catalysis, the enamine might be the major nascent product, with subsequent optimisation to the (2*R*)-imine and then (2*S*)-imines occurring in solution [7,9]. The formation of the (2*S*)-imine and, maybe, (2*R*)-imine can also occur at the active site in crystallo, at least as observed for the (2*S*)-imine in our VIM-2: biapenem work and with previous VIM-1: meropenem work [19].

Detailed spectroscopic analyses are required, both in solution and in crystals, to understand the precise nature of the reactions, including tautomerisations, that carbapenems undergo when bound to SBL and MBL active sites. It should also be noted that the observed product ratios are dependent on the efficiency of β-lactamase catalysis relative to that of tautomerisation, either in solution or in crystallo. Nonetheless, the combined results suggest that work aiming to prolong the lifetimes of the β-lactamase-bound carbapenem products is of interest with respect to developing improved carbapenems. This might be achieved either by the modification of the C2 carbapenem side chain or other group, in order to optimise the tautomerisation of the bound enamine tautomer to the (2*R*)-imine and potentially more stable (2*S*)-imine tautomer. The combined mechanistic studies with current clinically used carbapenems, including biapenem, however, suggest that the synthesis of new types of carbapenems may be required to achieve this.

## 4. Materials and Methods

### 4.1. Protein Purification and X-ray Crystallography

Recombinant forms of VIM-1, VIM-2, and KPC-2 were purified to near homogeneity (by SDS-PAGE analysis), as described [20,30,31]. VIM-2 (8.5 mg/mL) was crystallised using the vapour diffusion method under previously described conditions (Art Robbins low profile 96 well Intelli-plates) [20,30,31]. Crystallisation plates were set up using a PhoenixRE (ArtRobbins, Sunnyvale, CA, USA) robot with droplets of volume 200–300 nL with a 1:1, 1:2, and 2:1 protein: reservoir ratio equilibrated against 80 μL reservoir solution. VIM-2 crystals appeared after 24–48 h and were soaked with ~200–300 nL of 50 mM biapenem aqueous stock solution for 10 min. The crystal was harvested, then cryo-cooled using liquid nitrogen and sent for data collection at the Diamond Light Source.

### 4.2. Data Collection and Processing

Diffraction data were collected on a single crystal at 100K at Diamond Light Source beamline I04-1. Images were indexed and integrated using FastDP [32]; the structure was solved by molecular replacement with Phaser using a previously reported VIM-2 structure as a model (PDB file 4BZ3) [30,31,33,34]. A dataset was collected on a crystal soaked with biapenem for 10 min. After the first round of refinement, the mF_o_–DF_c_ map was consistent with the presence of (a) biapenem-derived ligand(s). The ligand was manually fitted using WinCOOT before the second round of refinement. Subsequent refinements were carried out using the program Phenix, with manual model building in WinCOOT [35,36,37]. After further refinement, the electron density maps implied the presence of two different imposed products. Trial refinements with the (2*S*)-imine, the (2*R*)-imine, or the enamine (independently); the two imines together at 50% occupancy; and the enamine with either the (2*S*)- or (2*R*)-imine at 50% occupancy were carried out. The best fit was obtained with the (2*S*)-imine and the enamine with 50% occupancy. The geometry restraints for the biapenem (2*S*)-imine and enamine-derived products were calculated using electronic ligand builder and optimisation workbench (eLBOW), and the electron density maps were calculated using PHENIX [35].

### 4.3. ^1^H NMR Studies

NMR-monitored assays were performed with VIM-1 (0.28 μM, final concentration) VIM-2 (0.28 μM), and KPC-2 (0.28 μM) mixed with 5 mM of biapenem in a 50 mM sodium phosphate solution at pH 7.6 with 10% (*v/v*) D_2_O, using trimethylsilylpropanoic acid as an internal standard (2 μL of a 3 mg/mL solution). Experiments were conducted using a Bruker AVIIIHD 600 MHz spectrometer equipped with a Prodigy broadband cryoprobe. Water signal suppression was conducted using excitation sculpting with perfect echo. Chemical shift assignments were based on reported values; data were processed using MestReNova [38]. Time-courses are presented with earliest time point at the bottom and last time point (40 min) at the top (Figure 2 and Appendix A). The spectrum of biapenem (5 mM) is displayed at the bottom. Each spectrum was acquired at 2.5 min intervals (16 scans). Note that the presence of the enamine product was assigned based on previous studies.

## Figures and Tables

**Figure 1 antibiotics-11-00396-f001:**
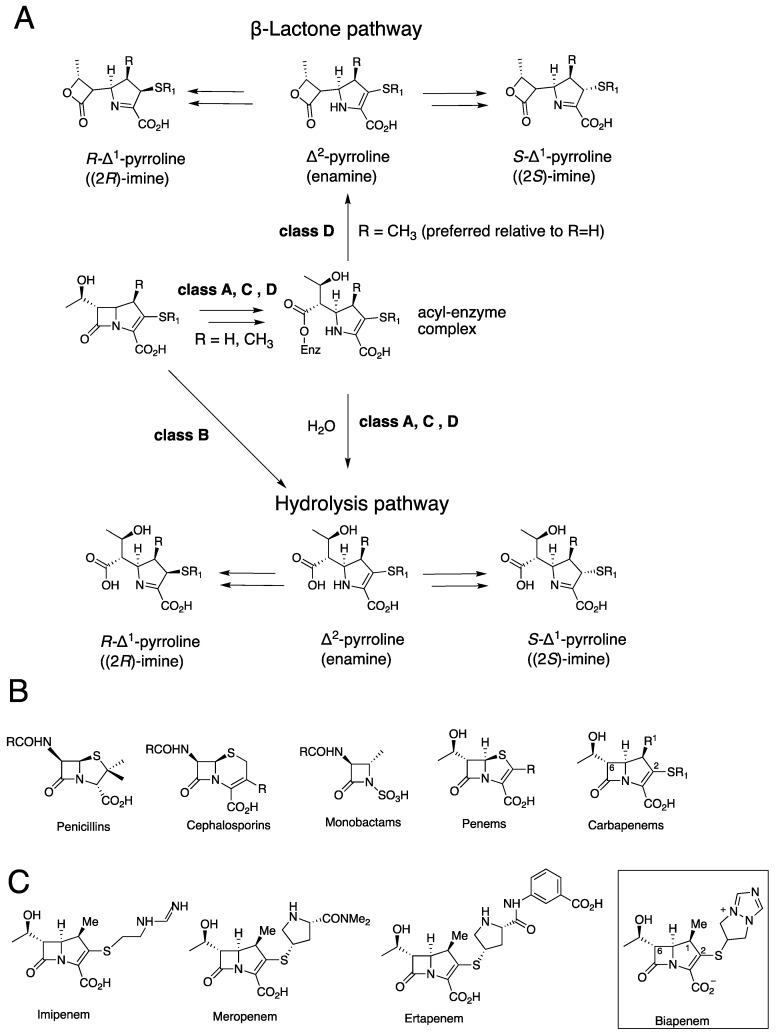
Carbapenem hydrolysis by β-lactamases. (**A**) It is proposed that SBLs and MBLs hydrolyse carbapenems to produce an enamine (Δ^2^-pyrroline), which during efficient catalysis in solution isomerises to give (2*R*)- and (2*S*)-Δ^1^-imine products. Both enamine and (2*S*)- and (2*R*)-imine carbapenem-derived ligands have been observed by crystallography at MBL/SBL/transpeptidase active sites. Lactones can also be formed, at least, in the case of class D SBLs. (**B**) Classes of clinically used β-lactam antibiotics (for carbapenems R^1^ = H or Me). (**C**) Examples of clinically used carbapenems; note the unusual positively charged bicyclic bicyclotriazolium C2 side chain of biapenem.

**Figure 2 antibiotics-11-00396-f002:**
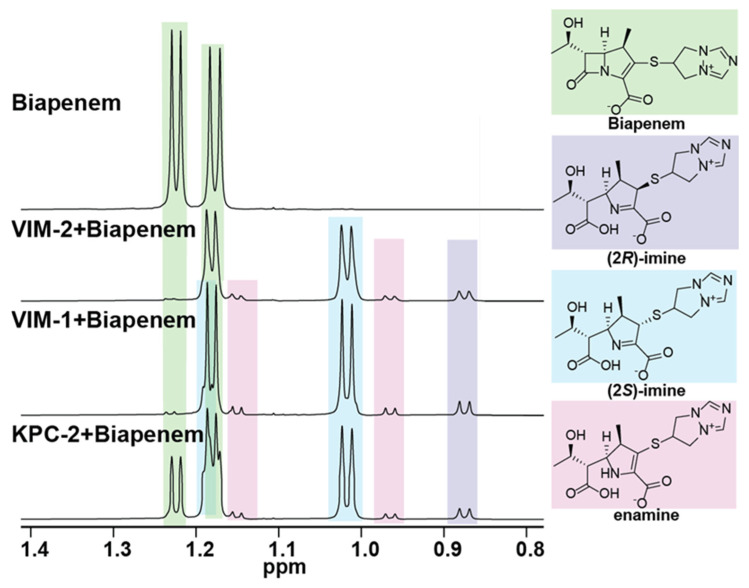
High-field regions of ^1^H NMR spectra (600 MHz) for reaction of biapenem with the MBLs VIM-1 or VIM-2, or the SBL KPC-2. Biapenem (5 mM, green) was treated with the purified β-lactamase (280 nM, 30 min) in 50 mM of sodium phosphate at pH 7.6 (with 10% *v/v* D_2_O). Formation of enamine (pink) and (2*R*)-imine and (2*S*)-imine (purple and blue, respectively) products was observed in all three cases. Time-course analyses imply the (2*S*)-imine is the major and ‘thermodynamic’ product (Appendix A). The slowest rate of biapenem turnover was observed with KPC-2 (Appendix A), with increasing rates observed for VIM-1 (Appendix A), and VIM-2 (Appendix A).

**Figure 3 antibiotics-11-00396-f003:**
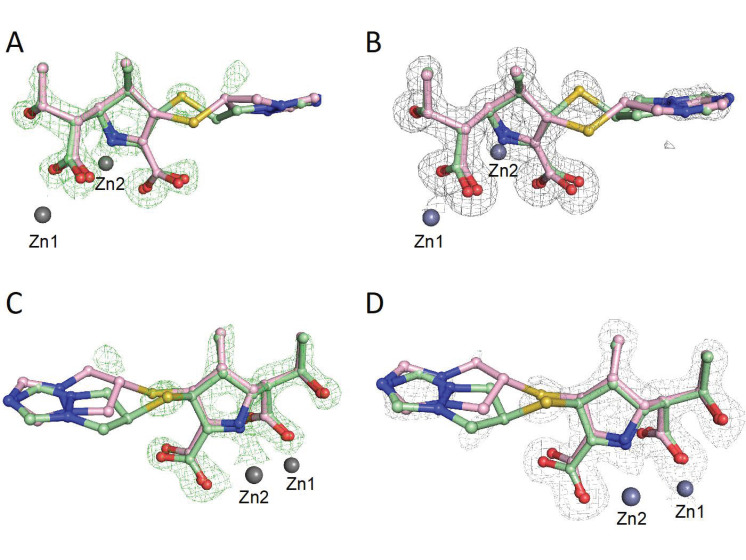
Electron density maps for biapenem-derived products in complex with VIM-2. (2*S*)-Imine: pale green balls and sticks; enamine: pink balls and sticks. (**A**,**C**) show an OMIT map (green mesh) of mFo-DFc contoured to 3σ (PDB: 6Y6J). (**B**,**D**) show the 2mFo-DFc difference map in grey mesh, contoured to 1σ. Note the weak density for the biapenem-derived bicyclotriazolium sidechain, indicating the presence of multiple conformations.

**Figure 4 antibiotics-11-00396-f004:**
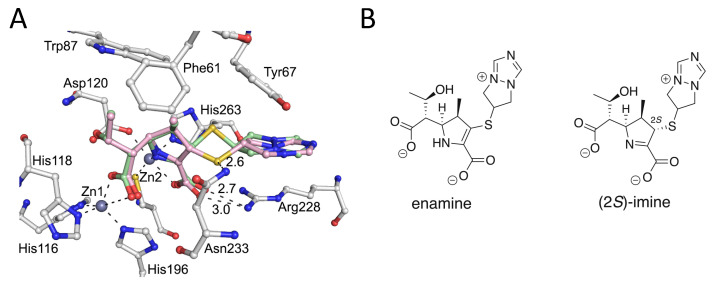
Views of a crystal structure of VIM-2 complexed with biapenem-derived ligands (PDB 6Y6J). (**A**) Interactions occurring between the (*2S*)-imine (green) and enamine (pink) products at the VIM-2 active site. Note that both products interact via their C3 carboxylate with Arg228 and that the sulfur of the thioether sidechain of the enamine is positioned adjacent to the primary amide of Asn233. (**B**) The (*2S*)-imine and enamine tautomers were each modelled at 50% occupancy.

## Data Availability

PDB available at https://www.rcsb.org/structure/6Y6J (accessed on 20 January 2022).

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
