# Peer review of "Studies on the Reactions of Biapenem with VIM Metallo β-Lactamases and the Serine β-Lactamase KPC-2"

_antibiotics, 2022, doi:10.3390/antibiotics11030396_

Round 1

Reviewer 1 Report

Review comments attached.

Author Response

 Figure 1C. Meropenem is misspelled.

            Thank you – corrected.

 Page 3 line 63. Currently reads as “Gram positive / negative bacteria as well as anaerobic/ aerobic bacteria”. The spacing/non-spacing in the use of the backslash (/) is inconsistent in this sentence and with other usage in the manuscript. Also, “Gram-positive” should be used in place of “Gram positive”.

            Thank you – corrected.

 Page 3 line 66. dihydropeptidase should be “dehydropeptidase”

            Thank you – corrected.

 Page 3 line 85. The parenthetical statement requires a space between 1.3 and ppm.

            Thank you – corrected.

 Page 5 line 114. Require a space between minutes and [17]

            Thank you – added.

 Page 5 line 116 The text indicates the diffraction data extended to 1.3 Å-resolution. However, the diffraction data and refined structure uses data only to 1.5 Å. The CC1/2 indicates extension to 1.3 Å is appropriate. Please correct this discrepancy by either changing the text in the body of the manuscript or by changing the data table to reflect reprocessing of the diffraction data and structural refinement to 1.3 Å-resolution.

            The data were indeed collected to 1.3 Å however due to poorer completeness in the highest resolution shell the resolution was cut off to 1.5 Å for data processing and refinement. We thank the reviewer for noticing this discrepancy and have clarified in the Main Text and SI.

 This is further highlighted by comparison with the correct drawing in panel C of the (2R)-imine, which shows those bonds in the cis configuration.

We thank the reviewer for spotting this error – they are absolutely correct, the correct imine stereochemistry is (2S), which matches that in solution.

Additionally, the chemical structure for the enamine in panel C is incorrect (the enamine in figure 2 is correct however). There should not be 2 chiral centers in the heterocycle (i.e. the bond linking the central heterocycle to the thioether should not be a wedge). This is an easy mistake to make and easy to fix. Please ensure that the chemical structures in the crystal structure correspond with the drawn chemical structures throughout the manuscript and the supplementary material

Thank you – this has been corrected and the chemical and crystallographic structures all correspond.

Also, it is not clear why the 2nd configuration is modeled as the enamine and not the other imine. It looks to my eye that the centroid of the sulfur peak in the map does not match very well with the modelled position of the enamine sulfur atom. Would the density better match the modeling of the other imine? Is the lifetime of the enamine known? It would seem to be much shorter than the combined lifetimes of the R and S imines. Maybe briefly revisit the structure to see if the 2 imines fit better than a combination of an enamine and imine.

Six trial refinements were made with both (2S)-and (2R)-imines alone, the two imines together and the enamine with either the (2S) or (2R)-imine. The best fit was clearly obtained with the (2S)-imine and the enamine, though we cannot rule out low levels of the presence of the (2R)-imine and other ligands (note in the NMR there were some unassigned peaks). The occupancy percentage is an approximation. We have made the procedure we sued clear in the revised manuscript. The assigned sulfur occupancy fit best with the enamine and (2S)-imine. We will work to obtain higher resolution MBL-carbapenem structures in future work. The idea to look at anomalous maps for sulfur is reasonable, but these will be very weak (and likely inconclusive) as dispersive differences very small (f’-f” = 0.0213e) at the wavelength (0.9159A) the data were collected at

Reviewer 2 Report

Title: Biapenem Hydrolysis by metallo and serine-β-lactamases 

This manuscript describes the products of the reaction of biapenem with several clinically relevant metallo- beta-lactamases (MBL) and serine beta-lactamases (SBL) as well as a crystal structure of a MBL with bound product. Overall, this is a well-written report with quality data and relatively straightforward interpretation. However, I believe the authors could expand on some aspects of this work, particularly the crystallography. Specific suggested changes are indicated below: 

  1. I think the description of the proteins of interest could be expanded a bit in the introduction. What is the prevalence of these proteins among antibiotic resistant bacteria? From what organisms are the proteins used in this study?
  2. In Figure 3C the bond to the sulfur atom of the enamine is drawn as if it were chiral, which it is not.
  3. The legend to Figure 3A indicates it shows “mFo-Fc” density. I am not familiar with this terminology. Is this a weighted 2Fo-Fc map? I would like to also see an omit map for the product, maybe in more than one orientation to get a better sense of the 3D arrangement of atoms. Also, I would be interested to see some of the alternate modeling refinements showing residual difference density in each case. This could go in supplemental data. 
  4. The introduction mentions the apparently important bicyclotriazolium moiety of biapenem. Does the crystal structure shed any light on its inhibitory function? It appears to have quite weak electron density, indicating multiple conformations. Interaction between it and protein residues also seems minimal, except perhaps for some pi-stacking (?). In any case, the authors might mention in the discussion what insights, if any, the structure provides for the special functions of biapenem.
  5. Please check the references section. There appear to be some formatting issues here including missing journal names, article titles, etc. 
